# Photosynthetic Activity Measured In Situ in Microalgae Cultures Grown in Pilot-Scale Raceway Ponds

**DOI:** 10.3390/plants13233376

**Published:** 2024-11-30

**Authors:** Jiří Masojídek, Karolína Štěrbová, Victor A. Robles Carnero, Giuseppe Torzillo, Cintia Gómez-Serrano, Bernardo Cicchi, João Artur Câmara Manoel, Ana Margarita Silva Benavides, Marta Barceló-Villalobos, Joaquín Pozo Dengra, Vince Ördög, Juan Luis Gómez Pinchetti, Francisco Gabriel Acién Fernándéz, Félix L. Figueroa

**Affiliations:** 1Laboratory of Algal Biotechnology, Centre ALGATECH, Institute of Microbiology, Czech Academy of Sciences, 37901 Třeboň, Czech Republic; sterbova@alga.cz (K.Š.); manoel@alga.cz (J.A.C.M.); 2Faculty of Science, University of South Bohemia, 37001 České Budějovice, Czech Republic; 3Institute of Blue Biotechnology and Development, Málaga University, (IBYDA), 29004 Málaga, Spain; victor_22891@hotmail.com; 4CNR—Institute of Bioeconomy, 50019 Sesto Fiorentino, FI, Italy; giuseppe.torzillo@cnr.it (G.T.); bernardo.cicchi@ibe.cnr.it (B.C.); 5CIMAR—Centro de Investigación en Ciencias del Mar y Limnología, Universidad de Costa Rica, San Pedro 11501, Costa Rica; margari50@gmail.com; 6Department of Chemical Engineering, University of Almería, 04001 Almería, Spain; cinti4201@hotmail.com (C.G.-S.); marta@gmail.com (M.B.-V.); facien@ual.es (F.G.A.F.); 7CIESOL and Solar Energy Research Centre, Joint Centre University of Almería-CIEMAT, 04001 Almería, Spain; 8BIORIZON BIOTECH, Parque Científico Tecnológico de Almería, 04120 Almería, Spain; jpozo@biorizon.es; 9Department of Plant Science, Faculty of Agricultural and Food Sciences, Széchenyi István University, 9200 Mosonmagyaróvár, Hungary; ordog.vince@sze.hu; 10Research Centre for Plant Growth and Development, School of Life Sciences, University of KwaZulu-Natal, Scottsville 3201, South Africa; 11Spanish Bank of Algae (BEA), Instituto de Oceanografía y Cambio Global (IOCAG), Universidad de Las Palmas de Gran Canaria, 35001 Las Palmas, Spain; juan.gomez@ulpgc.es

**Keywords:** green microalgae, chlorophyll (Chl) fluorescence, electron transport, non-photochemical dissipation, oxygen production, photosynthesis, photic layer, *Scenedesmus*, RWP

## Abstract

The microalga *Scenedesmus* sp. (Chlorophyceae) was cultured in a raceway pond (RWP) placed in a greenhouse. The objective of this case study was to monitor the photosynthesis performance and selected physicochemical variables (irradiance, temperature, dissolved oxygen concentration) of microalgae cultures in situ at various depths of RWP. The data of actual photochemical yield Y(II), the electron transport rate monitored by in vivo chlorophyll fluorescence and photosynthetic oxygen production both in situ and ex situ revealed that (i) even in diluted cultures (0.6 g DW L^−1^), the active photic layer in the culture was only about 1 cm, indicating that most of the culture was “photosynthetically” inactive; (ii) the mechanism of non-photochemical quenching may not be fast enough to respond once the cells move from the surface to the deeper layers; and (iii) even when cells were exposed to a high dissolved oxygen concentration of about 200% sat and higher, the cultures retained a relatively high Y(II) > 0.35 when monitored in situ. The presented work can be used as exemplary data to optimize the growth regime of microalgae cultures in large-scale RWPs by understanding the interplay between photosynthetic activity, culture depth and cell concentration.

## 1. Introduction

Two basic approaches are used for microalgae production in solar bioreactors: one possibility is cultivation in open systems (with direct interaction of the microalgae culture with the environment), while the other employs closed or semi-closed bioreactors with no direct contact between the culture and the outside environment [1,2]. At present, the most frequently used “open” cultivation systems for the commercial production of microalgae are various types of so-called RWPs due to their lower construction cost and easier maintenance compared to closed bioreactors. Early installations of RWPs were introduced in the 1950s–1960s by Oswald and co-workers [3,4] and their designs were advanced in the 1970s–1980s by the use of paddlewheel mixers which reduced the shearing forces on the cells and their energy requirements [5]; further improvements have been made ever since [6,7,8].

At present, the production of RWPs from hundreds to several thousand square meters has been reported [1]. The key construction parameters are the depth of the culture and the total area occupied. The area is usually divided into two or more channels along which the culture is recirculated by paddle wheels or impellers (see Figure 1a) that maintain a culture velocity of about 0.2–0.3 m s^−1^. The recommended length-to-width (L/W) ratio of the channels is about 15 [9]. The surface-to-total-volume ratio in these systems is about 5–10 m^−1^. Channel depth should be in the range of 0.2–0.3 m; it is recommended that they operate at a lower culture depth to increase light penetration, biomass concentration and the stability of the cultures. The shorter the light path, the higher the biomass density that can be maintained (refer to [1]).

However, the use of RWPs is restricted—due to the limited control of cultivation conditions and contamination—to a few “robust” and fast-growing microalgae genera (e.g., *Chlorella, Scenedesmus, Nannochloropsis*), those that are cultured ina selective medium (e.g., *Arthrospira*) or marine strains cultured in seawater (e.g., *Dunaliella* and *Nannochloropsis*). The cultures in RWPs are usually grown at low biomass densities, ranging between 0.5 and 1 g dry weight L^−1^ depending on pond depth, which in turn usually changes with the seasons. The cultures are usually operated in a semi-continuous regime, with cells collected every 4–5 days depending on the growth season and generally maintained at higher biomass densities to facilitate harvesting [10].

Available light is a key factor limiting the productivity of microalgae mass cultures outdoors [11]. However, the impact of irradiance on the productivity of outdoor microalgae cultures is rather complex due to the interaction with other environmental variables. The average amount of photon energy received by a single cell is a combination of several factors: light intensity, cell density, culture layer and the rate of mixing, as well as the construction parameters of cultivation units (for a review, see, e.g., [12,13,14]). In any case, the irradiance regime should be optimized, as excess light can damage the photosynthetic apparatus, particularly in combination with extreme temperatures or high oxygen concentrations [15,16].

Microalgae in outdoor mass cultures exposed to high ambient irradiance can generate high concentrations of dissolved oxygen (DO) due to photosynthetic activity. Though oxygen evolution is sometimes overlooked in large-scale units, high concentrations of DO in cultures, especially those occurring under high irradiance, can result in photoinhibition and photorespiration, which entail a reduction in photosynthetic activity and growth. In open units, the build-up of DO concentrations as much as 3–4 times higher than the air saturation values can be observed during the day, which may partly decrease the photochemical yield of microalgae cultures [17,18]. The maintenance of DO levels below critical concentration (about 200% of air saturation) often requires degassing or efficient culture mixing.

Improving culture productivity requires constant measurement of physicochemical variables, such as pH, temperature, DO concentration and nutrient levels, but most crucial is monitoring and optimizing photosynthetic performance. The primary indications of adverse cultivation conditions can be detected as a reduction in the photosynthetic activity of a microalgae culture, which subsequently slows down its growth and productivity [19]. Finding rapid, reliable and robust techniques to evaluate variations in microalgae activity has been one of the major tasks in the online monitoring of pilot and large-scale cultivation units [10,20,21]. Various monitoring methods have been used to adjust growth regimes, showing that the photosynthetic variables (oxygen production and respiration and Chl fluorescence variables, i.e., photochemical yield or the electron transport rate) monitored in situ/ex situ reflect the physiological status of the microalgae culture well and can provide primary information on photosynthetic activity, which is reflected in growth and biomass productivity [10,17,21,22,23].

The most reliable direct method of monitoring microalgae cultures is to follow the photosynthetic changes via in situ tracking of the actual situation. The other, more time-consuming, possibility is to assess the culture state ex situ using microalgae samples withdrawn from a cultivation unit. Some variables, such as photosynthetic oxygen production or PSII photochemical yield and electron transport rate, are used to correlate photosynthetic activity and growth [20,24,25].

In this case study, we aimed to survey the photosynthetic activity of outdoor microalgae cultures at various parts and depths of RWPs using several photosynthesis monitoring techniques. Particularly, cultures of the green microalga *Scenedesmus* sp. (further abbreviated as *Scenedesmus*) were examined. They are characterized by their robustness and fast growth. The photosynthetic activity of the cultures was measured in parallel with other variables (irradiance, temperature, dissolved oxygen concentration). The data showed that even in cultures that were not very dense (~0.6 g dry weight L^−1^), the active photic layer was limited to about 1 cm. Surprisingly, when the cells were moved to the bottom layer, i.e., to low light conditions, by mixing, the photosynthetic apparatus continued to operate at a lower efficiency as a result of some form of not fully diminished quenching, which could reduce photochemical performance, particularly at midday, when irradiance is higher. The presented work can be used as exemplary data to optimize the growth conditions of microalgae cultures in large-scale RWPs.

## 2. Results

### 2.1. Culture Growth, Irradiance and Oxygen Production

Five-day trials were carried out on sunny and warm days in mid-summer (August). A culture of the green microalga *Scenedesmus* sp. was grown in an RWP with an area of 76 m^2^, in which three measuring sites were defined (1—after the sump; 2—at the first bend; 3—close to the end of the second channel before deflectors) (Figure 1). The choice of the measurement points included sites where the mixing was less efficient, such as at the bends, or where it was expected to be considerably higher, such as after the paddlewheels and the sump.

The biomass density of the *Scenedesmus* culture was about 0.6 g L^−1^. The culture depth was about 14 cm. As concerns the temperature and DO concentration changes in the *Scenedesmus* culture, the data varied between 27.5–28 °C in the morning (9:00 h) and 35–36 °C in the afternoon (17:00 h) (Figure 2a—values are a range of data recorded in three measuring positions). DO concentration values were found to be between 26 and 78%sat in the morning, increased to 171–258%sat at midday (13:00 h) and ranged between 178 and 258%sat at 17:00 h (Figure 2b—values represent a mean of data recorded in five measuring positions). The Chl content remained rather stable during the measurement, between 12 and 15 mg L^−1^ (Figure 2c). The Chl concentration varied on day 1 and day 2 as it slightly (statistically) increased from the morning to the afternoon. On day 3, it did not change, since the culture was growing slowly and the Chl concentration remained stable.

The typical daily course of irradiance measured in the greenhouse indicated dawn at about 6:30 h and dusk at about 21:00 h; the range of the highest irradiance was seen between 14:00 h and 15:00 h (Figure 3). The ambient irradiance maxima of about 1800 µmol photons m^−2^ s^−1^ measured at 13:00 h were usually 10–15% higher than those inside the greenhouse. Therefore, the irradiance intensities measured at the surface of the RWP during the trial were about 300, 1350, 1600 and 1100 µmol photons m^−2^ s^−1^ at 9:00, 11:00, 13:00 and 17:00 h, respectively. The irradiance intensities measured in situ in the cultures at the depths of 1, 4 and 7 cm were in ranges of 85–250, 20–85 and 6–21 µmol photons m^−2^ s^−1^, respectively (Figure 3). The data showed that there was a dramatic difference between the irradiance intensity at the surface and at 1 cm culture depth (80–90% decrease) and then another 50% decrease at depths between 1 and 4 cm. Irradiance intensity at 7 cm depth was rather low and this depth is probably outside the photic zone of the culture.

### 2.2. Photosynthesis Monitoring In Situ

Values of the actual photochemical yield Y(II) were measured in situ in the culture at two depths (0.6 and 4 cm) during the three-day trials (Figure 4). The culture depths for measurement were selected to provide an average light intensity (midday) of 150–200 µmol photons m^−2^ s^−1^ at 0.6 cm depth (close to saturating irradiance for growth at this biomass density) and about 20 µmol photons m^−2^ s^−1^ at 4 cm (low irradiance close to the photosynthesis compensation point). It is important to note that the data presented here are a mean of data measured at three positions in the RWP (see Figure 1b) and there was not much variation among the values measured at various positions. The Y(II) values were monitored at 9:00, 13:00 and 17:00 h on days 1, 2 and 3. In this trial, the highest values of Y(II), between 0.49 and 0.71, were found at 9:00 h (Figure 4a–c); there was not much difference between the values measured at both depths due to low ambient irradiance in the morning (Figure 3b). At 13:00 h and 17:00 h, the Y(II) values measured at 0.6 cm and 4 cm depths were lower than in the morning, between 0.31 and 0.51, due to higher ambient irradiance penetrating the culture, which caused a higher PQ pool reduction (see Section 2.3). Even at 4 cm depth, cells preserved some form of quenching developed on the surface, which could further reduce ETR capacity, particularly at midday, when irradiance intensity is higher.

The data on the ambient irradiance (PAR) penetrating inside the culture and the actual PSII photochemical yield Y(II) were measured in situ/online during the diurnal cycle using submerged irradiance and fluorescence sensors, placed close to each other, which were connected to a Junior-PAM fluorimeter. Then, the relative electron transport rate (rETR) was estimated [Y(II) × E_PAR_]. The irradiance intensity started to rise between 8:00 and 9:00 h and it diminished at about 21:00 h, which means that the culture was exposed to ambient light for about 12–13 h (Figure 5). As mentioned above, there was not much variation among the values of actual photochemical yield Y(II) measured at various positions (Figure 4); thus, only one position, point 1 in the RWP, downstream of the sump, was selected for online measurements at two depths, 0.6 and 4 cm. An irradiance intensity of about 150–200 µmol photons m^−2^ s^−1^ was recorded at 0.6 cm depth between 14:00 and 17:00 h (Figure 5a). In this period, the maximum values of rETR were also estimated and were between 115 and 160 µmol e^−^ m^−2^ s^−1^; these values followed a course corresponding to the irradiance intensities measured inside the culture. The course of Y(II) values was antiparallel to irradiance due to the degree of PSII reduction—in light periods, it was between 0.20 and 0.45, while in dark periods, it increased up to 0.6. When the irradiance inside the culture was measured at 4 cm depth, much lower maxima of irradiance intensities—between 15 and 20 µmol photons m^−2^ s^−1^—were found (Figure 5b). There was not much difference in Y(II) values between the night and day periods, as the penetrating irradiance was low; the trend only showed a slight decrease after sunset (Figure 5b). Thus, the estimated rETR at 4 cm depth showed maximum values between 10 and 20 µmol e^−^ m^−2^ s^−1^, which was less than one-tenth of those found at 0.6 cm depth.

### 2.3. Photosynthesis Measurements Ex Situ

The maximum photochemical yield (Fv/Fm), the maximum relative electron transport rate (rETR_max_), the maximum non-photochemical quenching (NPQ_max_), the maximum rate of photosynthetic oxygen production (P_max_), the maximum rate of dark respiration (Resp) and the fraction of respiration in gross photosynthesis (Resp/PS_gross_) (in %; PS_gross_ is the sum of P_max_ + Resp) were measured ex situ using culture samples taken from the RWP at particular daytimes, namely, 9:00, 13:00 and 17:00 h (Figure 6). The values of Fv/Fm, rETR_max_ and NPQ_max_ were calculated from light–response curves (LRCs) of Chl fluorescence, while P_max_, Resp and Resp/PS_gross_ were determined from LRCs of oxygen production/consumption. Both LRCs were measured in parallel.

The values of Fv/Fm ranged between 0.63 and 0.79 and were relatively aligned (Figure 6a). rETR_max_ values were found to be between 110 and 150 µmol e^−^ m^−2^ s^−1^ (Figure 6b), while the NPQ_max_ was between 0.25 and 0.63, and the values decreased from day 1 to day 3; the lowest values were obtained on day 3 (Figure 6c). The ranges of P_max_ and Resp were found to be between 9 and 10.8 pmol O_2_ cell^−1^ s^−1^ and 0.48–1.6 pmol O_2_ cell^−1^ s^−1^, respectively (Figure 6d,e). The data on the ratio of Resp/PS_gross_ revealed that the minimum values, between 4.6 and 9.8%, were achieved in the morning, while the highest value—between 17.8 and 28.9 pmol O_2_ m^−2^ s^−1^—was found in the afternoon (Figure 6f). These data confirmed an increasing trend from the morning to the afternoon and also from day 1 to day 3. The highest values, between 11 and 15.4, were found on days 2 and 3 at 13:00 and 17:00 h. Considering the values of Pmax and Resp, it was obvious that the ratio of Resp/PS_gross_ showed an increasing trend.

Culture samples taken from the RWP at 9:00, 13:00 and 17:00 h on days 1–3 were also examined using fast fluorescence induction kinetics (OJIP test) to estimate the redox status of quinone electron acceptors in the PSII complex (Figure 7). Examples of the fluorescence induction curves were recorded at 13:00 h for all three days (Figure 7a). The curves were analyzed to evaluate the reduction status of the PSII electron acceptors. From the fluorescence levels at points J and I, variables Vj and Vi were calculated. These variables express the redox status of quinone electron acceptors (see Methods). The trends in Vj and Vi were similar in the *Scenedesmus* culture (Figure 7b). The data showed that on day 3, the Vi values increased by about 10%, which might indicate a higher reduction rate of the PQ pool in the PSII complex compared to day 1.

## 3. Discussion

In full sunlight, microalgae cells receive more light energy for most of the day than they can use in photosynthesis. As a result, even under optimal conditions, the photosynthetic efficiency of microalgae cells is less than one-third of their theoretical maximum (10% of solar light). If this excess energy is not dissipated, then the formation of highly reduced electron carriers in the photosynthetic electron transport chain will occur, entailing the formation of harmful reactive oxygen species (ROS) [26]. In outdoor cultures, microalgae cells are subjected to fast changes in light intensity varying within seconds due to mixing [13]. Mechanisms collectively referred to as non-photochemical quenching (NPQ) are developed in higher plants [27] as well as microalgae [28] to dissipate excess energy as heat or to distribute excitation energy (state transition) between the two photosystems, thus protecting the photosynthetic apparatus from such damage [29]. The most significant component of NPQ is energy-dependent quenching (qE). According to the current understanding, qE is developed based on the synergistic action of lumen pH, xanthophyll interconversion and conformational changes in PSII antenna proteins Lhcb and PsbS protein [30]. Recently, the NPQ scenario was demonstrated in higher plants as NPQ = ΔpH + PsbS + LHCII [31]. It was proposed that NPQ is triggered by ΔpH either directly by the protonation of antenna components or indirectly by the activity of the xanthophyll cycle. LHCII complexes CP26 and CP29 can accept protons attaining high levels of quenching and are enriched in xanthophyll cycle carotenoids. A component that plays a crucial role in enabling the rapidly reversible component of NPQ, qE, is the PsbS protein, which acts like a switch that is also triggered by ΔpH and is localized closer to the LHCII antenna, thus being able to prompt it into a quenching state or make it sensitive to protonation [32].

In the culture of microalgae, like in higher plants, qE is activated when the amount of incoming light energy exceeds the capacity of electron sinks. The participation of another component, state transition quenching (qT), cannot be ruled out. The threshold light level inducing this process is lowered under unfavorable conditions, such as high salinity, nutrient deficiency or suboptimal temperature, which further limits photosynthesis [29]. While the requirements for NPQ development in green microalgae, i.e., the xanthophyll cycle, the proton gradient and LHCII aggregation, are comparable to those of higher plants, there also exist important differences [33]. The most prominent difference between higher plants and green microalgae in terms of their requirements for NPQ is that in several strains, the PsbS protein is not present and is replaced by the so-called Lhcsr proteins, ancient members of the LHC protein superfamily. It has been shown that the Lhcsr3 protein of *Chlamydomonas reinhardtii* has features which are comparable to those of PsbS [34]. Like this protein, Lhcsr3 is also sensitive to increases in proton concentration during the establishment of the photosynthetic proton gradient [34,35,36].

In this study, online measurements of the actual quantum performed in the *Scenedesmus* cultures showed that measurements performed at 0.6 and 4 cm depths followed the pattern of daylight independently of the culture layer. Due to the remarkable light extinction at 4 cm depth (about 20 µmol photons m^−2^ s^−1^ at midday, see Figure 5), one would expect that the actual quantum yield Y(II) would fully recover from transient forms of quenching (qE). On the contrary, the changes in Y(II) showed a pattern similar to that observed on the surface of the culture, i.e., antiparallel to light intensity. This behavior confirmed our previous observations showing that the relaxation mechanism of non-photochemical quenching may not be fast enough to facilitate the relaxation of the photosynthetic apparatus once the cells return to the deeper layers [28]. Such a situation may cause a significant loss of productivity, since some components of NPQ (e.g., photoinhibitory qI) still remain active during the transition of cells from the light-saturated layer to the light-limited part of the culture layer. Therefore, the rate of conversion of zeaxanthin to violaxanthin in the transition from sun to shade can accelerate NPQ relaxation, which can represent an important strategy to increase microalgae productivity [37]. Contrary to this, Y(II) decreased during the cultivation trial. The culture was set up on day 1, which might have induced a high morning value. Indeed, one should expect an increase in Y(II) or a stabilization with an increase in cell density. On days 2 and 3, the afternoon decrease in Y(II) can probably be ascribed to the synergistic effect of higher irradiance, excessive temperature and DO concentration. A similar effect was noted in the rise of the respiration/photosynthesis ratio measured ex situ (Figure 6f). Surprisingly, NPQ showed a tendency to decrease (Figure 6c). The reason for the decrease in NPQ during the cultivation trial is not clear; however, a stress effect of excessive temperature and dissolved oxygen concentration on the capacity to dissipate energy via NPQ cannot be ruled out [38]. This may occur during long-term adaptation of the photosynthetic apparatus to light and growth conditions through antenna proteins and carotenoid accumulation, changes in antenna size, PSII-to-PSI ratios, thylakoid-related architecture and others [31].

The presented data of photosynthetic activity indicate that only the photic layer close to the culture surface can substantially contribute to productivity in a relatively deep RWP, while deeper layers are photo-limited and are not photosynthesizing due to sub-saturating irradiance intensities (Figure 5). These findings are supported by previous measurements in green microalgae cultures grown in outdoor thin-layer cascades and RWPs [2]. Of course, we have to keep in mind that the cells are fluctuating due to mixing and there is an exchange in the populations along the culture column.

Photosynthetic activity affects dissolved oxygen concentrations, which usually reflect diurnal changes in irradiance intensity, showing a build-up from morning minima to maxima at midday and then declining in value through the afternoon. Under high photosynthesis rates, DO concentration can reach up to 25–30 mg L^−1^ (200–400% of air saturation) in the top layer of microalgae cultures, even in open ponds [18,23]. High DO concentrations during the day might have a certain impact on growth due to the potential slowdown of photochemical yield [17]. In the present trial, a significant build-up of DO concentration (>200% sat) was observed, which indicated that both cultures were photosynthetically active (Figure 2b). Nevertheless, the data showed that in deep RWPs, degassing is not efficient enough to counteract oxygen accumulation in culture volumes (Figure 2). Dark respiration, photorespiration and also the Mehler reaction related to PSI may contribute to reducing the photoxidation damage [39,40,41]. Environmental constraints such as photo-stress, high temperature, drought or high salinity stimulate the activity of alternative PS I-driven electron transport, providing additional flexibility to protect against unfavorable conditions [42]. Below the light saturation point of photosynthesis, the ratio between ETR and PS_gross_, measured as oxygen production, was found to be close to the theoretical value of 5, whereas at higher irradiances, the ratio was 7–15, indicating that the process of oxygen consumption had an important role, in addition to respiration as photorespiration and the Mehler reaction [20]. An ETR/PS_gross_ ratio higher than 5 has been related to stressful conditions, excess of light or nutrient limitations [43].

In this study, online measurements of the actual quantum yield Y(II) performed in cultures of *Scenedesmus* at 4 cm depth showed a diurnal pattern similar to that observed close to the surface of the culture, i.e., antiparallel to light intensity, as the minimum was recorded at midday (Figure 5). Due to light extinction, one might expect that the Y(II) yield would be higher in deeper layers (i.e., recovered via PSII reoxidation), but the cultures are mixed and the cells probably come from upper, more active (illuminated) layers. If the courses of Y(II) and DO concentration are compared (Figure 2 and Figure 5), a certain decrease in Y(II) is assumed to occur in the afternoon due to high DO concentrations of about 300%sat, as was found previously [16]. In the present trial, even when DO concentrations were quite high (Figure 2b), this did not have a considerable effect on photosynthetic activity (Figure 4c). This means that the culture was still photosynthesizing, although its activity was lower (Figure 5).

Further comparison of photosynthetic variables [Fv/Fm, Y(II), rETR_max_, P_max_, Resp] measured ex situ in samples taken from the cultures revealed that the *Scenedesmus* culture was photosynthetically active (Figure 5, Figure 6 and Figure 7). High P_max_ photosynthesis rates were associated with higher respiration rates in the *Scenedesmus* cultures (Figure 6d,e). The Resp/PS_gross_ ratios may have very different performances during the diurnal cycle, since both variables are substantially modified.

The maxima of the rETR_max_ values measured in situ and ex situ were similar (115–160 µmol e^−^ m^−2^ s^−1^) (Figure 5a vs. Figure 6b). Generally, in situ outdoor measurements of ETR are carried out under natural conditions (i.e., irradiance, temperature), monitor the actual situation in microalgae cultures and usually show higher activities. These activities are usually higher than those measured ex situ in samples taken from outdoor units due to their ‘light’ history.

The kinetics of fast fluorescence induction did not show any deep disturbance in the PSII complex (Figure 7). This suggests that electron transport on the acceptor side of the PSII complex was slightly affected, and that it was only delayed, probably due to a partial over-reduction in Q_A_. A certain increase in Vj and Vi values was observed on day 3, which might suggest that electron transport in the cell was slowed down. This shows that the electron transport on the acceptor side of the PSII complex was not affected as much. Some increase in Vj and Vi values was observed on day 3, which might suggest that the electron transport in the cell was slowed down.

The changes in non-photochemical quenching (NPQ_max_) and Resp in both cultures (Figure 6c,e) showed that the values had an antiparallel course. This indicates that the part of light energy absorbed by the photosynthetic apparatus may be dissipated via non-photochemical quenching or respiration [26,28]. The relaxation mechanism of NPQ may not be fast enough to relax the photosynthetic apparatus. This mechanism still remained partially active during the transition of the cells from the light-saturated layer on the surface of the culture to the light-limited zone. An accelerated rate of NPQ relaxation might be related to a decrease in luminal pH, inducing xanthophyll cycle activation during the transition from sun to shade, which can be an important strategy for maintaining microalgae growth [28,37]. Situations of intense light dissipation can occur frequently in mass cultures of microalgae due to excessive biomass harvesting (dilution), which exposes cells to oversaturating irradiance for some time [13].

In this work, non-photochemical quenching was calculated as NPQ = (Fm − Fm’)/Fm’ [44]. In some reports, NPQ is also expressed as the ratio of the two components, i.e., Φ(NPQ)/Φ(NO) [45,46] or Y(NPQ)/Y(NO) [47]. These two components of non-photochemical quenching can be used to estimate the fluxes of excitation energy into competing pathways. Y(NPQ) should correspond to the fraction of energy dissipated in the form of heat via the down-regulatory photoprotective mechanism with the participation of carotenoids, while Y(NO), according to the above-cited authors, reflects the fraction of energy that is passively dissipated in the form of heat and fluorescence, mainly due to closed (PQ-reduced) PSII reaction centers. The sum of the two components, i.e., Y(NPQ) + Y(NO), plus the actual photochemical yield Y(II) is 1. The lower values of NPQ indicate that non-regulated loss of energy (Y(NO)) dominates compared to the photoregulated energy dissipation mechanism (Y(NPQ)) related to carotenoids [46,48].

## 4. Materials and Methods

### 4.1. Plant Material and Cultivation

The culture of the green microalga *Scenedesmus* sp. (class Chlorophyta; the strain was obtained from the University of Almería, Spain) was grown phototrophically (in the form of four-cell coenobia; average size: 6 × 13 µm) in an inorganic medium (0.9 g L^−1^ NaNO_3_, 0.14 g L^−1^ KH_2_PO_4_, 0.18 g L^−1^ MgSO4 and 0.015 g L^−1^ commercially available mixture of micronutrients @karentol containing Fe, Cu, Mn, B, Zn, Mo and others in the form of chelate) in an open cultivation system, i.e., aRWP (Figure 1). The 75 m^2^ unit was placed in a greenhouse which was located at the facilities of the company Biorizon Biotech in Almería, Spain (GPS coordinates: 36°49′59.7″ N, 2°24′22.7″ W). The average depth of the RWP was about 14 cm (resulting in a surface-to-volume ratio of about 8) and a culture flow rate of about 0.3 m s^−1^ was maintained by rotating paddle wheels. Gaseous carbon dioxide (CO_2_) was supplied based on a pH-stat system to keep the value close to about 8.5. Flow deflectors were placed at both ends of the RWP. Downstream of the paddle wheel, there was a sump where CO_2_ could be added to the culture or aerated to remove oxygen [1]. A semi-continuous cultivation regime was used in this three-day trial; culture dilution was always performed in the morning, using a rate of 0.2 d^−1^. From the morning value of 0.6, the culture usually reached about 0.7 at 17:00 h.

### 4.2. Photosynthesis Monitoring

The photosynthetic activity of the microalgae populations was monitored by in vivo Chl fluorescence and oxygen production measurements directly in the RWP in situ, as well as ex situ in microalgae samples taken from outdoor cultures. The time in the figures corresponds to CEST (GMT + 1).

#### 4.2.1. In Situ Measurements

A fluorimeter (Mini-PAM, H. Walz GmbH, Effeltrich, Germany with red 650 nm measuring light) was used to monitor fluorescence in situ at two culture depths (about 0.6 cm and about 4 cm) at 3 positions (see Figure 1) and various daytimes—9:00, 13:00 and 17:00 h—using submergible light and fluorescence sensors.

Chl fluorescence data were also recorded in vivo/online in microalgae cultures during the diurnal cycle using four portable fluorimeters (Junior-PAM, H. Walz GmbH, Effeltrich) controlled via a USB interface by the WinControl-3 software, version 3.22, which was also used for data acquisition [10,25,26]. The fluorimeter was fitted with blue-light-emitting diodes (LED; 450–470 nm) to apply the measuring and saturating pulses. Ambient irradiance was used as actinic light. To estimate photosynthesis variables, a fluorescence light guide represented by a plastic filament (1.5 mm in diameter and with a length of 100 cm) and a spherical irradiance mini-sensor (US-SQS, H. Walz GmbH, Germany) were submerged into the culture in the RWP at point 1 (downstream of the sump) at a depth of about 0.6 or 4 cm. Photosynthetically active radiation (E_PAR_) (400–700 nm) and the actual (or effective) quantum yield of PSII (Y(II)) [=(Fm’ − F’)/Fm’] were measured at each depth at 10 min intervals; the variable F’ is the steady-state fluorescence level and Fm’ is the maximal fluorescence induced by a saturating light pulse [10,26]. The relative electron transport rate rETR = [Y(II) × E_PAR_] through PSII (µmol e^−^ m^−2^ s^−1^) was used to estimate photosynthetic performance, where E_PAR_ is the particular irradiance intensity measured in the culture (µmol photons m^−2^ s^−1^). rETR can adequately be used to follow the diurnal changes in photosynthetic activity (and physiological conditions) in outdoor microalgae cultures and can be used as a comparative variable at various points during the daytime in situ and ex situ.

#### 4.2.2. Irradiance, Temperature, Ph, Cell Counting and Dissolved Oxygen Measurements

Ambient irradiance (PAR) was measured as 10 s averaged values using a portable light meter (model LI-250A, LI-COR Biosciences, Lincoln, NE, USA) coupled with a flat quantum sensor (LI-190SA, cosine-corrected up to 80° angle of incidence; LI-COR Biosciences, Lincoln, NE, USA) or with a spherical light mini-sensor (US-SQS, H. Walz GmbH, Effeltrich, Germany). The values of culture temperature were measured at several positions (using an oximeter) and the values were averaged (Figure 2a). Dissolved oxygen (DO) concentrations were recorded by a hand-held oximeter (model Oxi 330, WTW GmbH, Weilheim, Germany) with temperature compensation. The data were estimated in % of saturation (%sat) (Figure 2b). For cell counts, the culture samples were fixed by 2.5% glutaraldehyde and diluted and numbers were determined using a cell counter (Multisizer 4, Beckman Coulter Inc., Brea, CA, USA).

### 4.3. Ex Situ Measurements

For ex situ measurements of in vivo Chl fluorescence and oxygen production, mixed microalgae samples (4 cm depth) were taken from outdoor cultures at specified times in the day (9:00, 13:00 and 17:00 h) as described previously [13,17].

#### 4.3.1. Photosynthesis Light–Response Curves

The photosynthetic light–response curves (LRCs; photosynthesis vs. irradiance dependency) of electron transport or oxygen production were measured in parallel using a pulse-amplitude-modulation fluorometer (PAM-2100, H. Walz, Germany using red 650 nm measuring and actinic light) and an oxygen-monitoring system (Oxylab+, Hansatech Instr. Ltd., UK) connected to a temperature-controlled chamber (DW2/2, Hansatech Instrument Ltd., Norfolk, UK). The curves were recorded in the samples after 10 to 15 min of dark adaptation at 8 red light intensities between 0 and 1800 µmol photons m^−2^ s^−1^. They were exposed to each intensity for 2 min at the set temperature according to the actual culture value. The minimum and maximum fluorescence levels (F_0_ and Fm; F_0_, basal fluorescence from fully oxidized reaction centers of PSII; Fm, maximum fluorescence from fully reduced PSII reaction centers) were determined in the dark-adapted samples. The maximum quantum yield was calculated as the ratio of variable and maximum fluorescence, Fv/Fm = (Fm − F_0_)/Fm, which expresses the maximum quantum yield of primary photochemistry [26]. The variable called the relative electron transport rate through PSII (rETR) was calculated as the product of the actual photochemical yield (Y(II)) multiplied by photosynthetically active radiation, as mentioned above (see, e.g., [49,50,51]). The values of rETR_max_ were calculated at the maxima of LRCs. Non-photochemical quenching (NPQ [=(Fm − Fm’)/Fm’]) was used to estimate non-photochemical energy dissipation [44].

In addition, rates of dark respiration (Resp) and photosynthetic oxygen evolution (PS) were measured as a function of irradiance in microalgae samples using a Clark-type oxygen electrode (Oxylab+ monitoring system, Hansatech Instr. Ltd., UK) mounted in a temperature-controlled chamber (DW2/2 chamber, Hansatech Instrument Ltd., UK) connected to a programmable light source to adjust the intensity increase in six 2 min steps between 0 and 950 µmol photons m^−2^ s^−1^ (saturating intensity for photosynthesis) [37]. The values of respiration and oxygen production are expressed in pmol O_2_ cell^−1^ s^−1^.

#### 4.3.2. Fast Fluorescence Induction Kinetics (Kautsky Curve or OJIP Test)

While the pulse-amplitude-modulation (PAM) technique gives information on the energy distribution between the photochemical and non-photochemical processes in photosynthesis, fast fluorescence induction kinetics provides information on the redox status of the electron transport chain in the PSII complex. The fluorescence induction curves were measured ex situ by a portable fluorimeter (AquaPen AP-100, P.S.I. Ltd., Brno, Czech Republic) in samples taken from outdoor cultures and dark-adapted for 10–15 min as described previously [22]. The curves of fast fluorescence induction kinetics were measured in a time range between 50 µs and 1 s, when the signal rises rapidly from the origin (O) to the highest peak (P) via two inflections, J and I [52]. The O point (50 µs) of the fluorescence induction curve represents a minimum value (designated as a constant fluorescence yield, F_0_) when the PQ electron acceptors (Q_A_ and Q_B_) of the PSII complex are oxidized. Inflection J occurs after ~2–3 ms of illumination and reflects a dynamic equilibrium (quasi-steady state) between Q_A_ and Q_B_. The J–I phase (at 30–50 ms) is due to the closure of the remaining centers, and I–P (ending at about 300–500 ms) corresponds to a full reduction in the plastoquinone pool (equivalent to the maximum fluorescence level, Fm) [52]. From the fluorescence levels at points J and I, variables Vj and Vi were calculated as follows: Vj = (F_2ms_ − F_0_)/(Fm − F_0_) and Vi = (F_30ms_ − F_0_)/(Fm − F_0_). These variables showed the redox status of quinone electron acceptors.

### 4.4. Analytical Measurements

The measurement of biomass content was carried out as dry weight (DW) determination by filtering culture samples on pre-weighed glass microfiber filters (GC-50) as described previously [53]. The filters with the biomass were washed twice with deionized water and dried in an oven at 105 °C for 8 h; they were then weighed (precision of ±0.01 mg) and the biomass amount was calculated.

Chl concentration was determined spectrophotometrically in methanol extracts (Figure 2c). The cells were collected by centrifugation and the pellets were resuspended in 100% methanol. Sea sand or glass beads were added, and the tubes were put into a laboratory ultrasound bath heated to 40 °C for 2 min and then cooled down in an ice bath and centrifuged. The absorbance of the supernatant was measured using a high-resolution spectrophotometer (model UV-2600, Shimadzu Corp., Kyoto, Japan) and the concentration of Chl was calculated according to Wellburn [54].

### 4.5. Statistical Analysis

Most measurements were performed in triplicate (n = 3); the means and standard deviations (SDs) are reported in the figures. Sigma Plot 11.0 (Systat Software Inc., San Jose, CA, USA) was used to determine significant differences between treatments. One-way analysis of variance (ANOVA and the Holm–Sidac test) using SigmaStat (Systat Software Inc., San Jose, CA, USA) were conducted for the comparison of variables in the trials. For Figure 2, Figure 4 and Figure 6, statistical differences between samples collected at the same time and at different sampling positions were studied. *p* values lower than 0.05 (*p* < 0.05) were considered to be significantly different. In graphs, the mean values designated by the same letter do not differ from each other, while different letters mean significant differences.

## 5. Conclusions

Photosynthesis-measuring techniques using Chl fluorescence both in situ and ex situ prove useful for culture monitoring, as they reliably reflect the physiological status of microalgae cultures and can be used to adjust suitable growth regimes.By illustrating the interplay between culture depth and cell concentration, the presented data can be used to optimize the growth of microalgae cultures in large-scale open pondss.Online measurements of oxygen production and fluorescence variables in various culture depths showed that the photic zone contributing to growth in an RWP is relatively thin, about 1 cm, and if this is the case, it comprises less than 10% of the volume, meaning that most of the culture is “photosynthetically” in the dark.The mechanism of non-photochemical quenching may not be fast enough to relax the photosynthetic apparatus once the cells return from the surface to the deeper layers.In the presented trial, the values of the maximum photochemical yield of PSII, Fv/Fm, were found to be between 0.62 and 0.79, which indicates that microalgae cultures might be mildly constrained at certain times during the day.The results also revealed that part of the light energy harvested by the photosynthetic apparatus is dissipated via non-photochemical quenching or respiration in response to variable environmental conditions.

## Figures and Tables

**Figure 1 plants-13-03376-f001:**
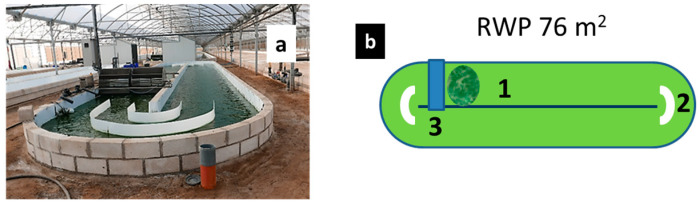
Picture and schematic diagram of the 76 m^2^ RWP used in this study. (**a**) The RWP was placed in a greenhouse which was located at the facilities of the company Biorizon Biotech in Almera, Spain (GPS coordinates: 36°49′59.7″ N, 2°24′22.7″ W). (**b**) Schematic diagram of the RWP with numbered positions where culture variables were monitored. Three measuring positions were defined (1—after the paddlewheel and sump; 2—at the first bend; 3—close to the end of the second channel before the deflectors and paddlewheel).

**Figure 2 plants-13-03376-f002:**
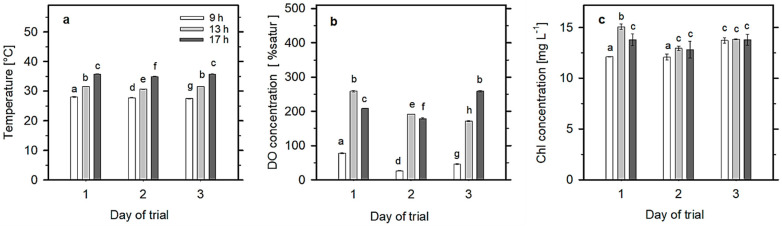
(**a**) Daily changes in temperature (°C), (**b**) dissolved oxygen concentration (saturation in %) and (**c**) Chl concentration measured in the cultures during the three-day trial. Data were monitored at 9:00, 13:00 and 17:00 h. Values are presented as a mean of figures measured at 3 positions in the RWP (see Figure 1) ± SD; those designated by the same letter a–h did not differ significantly from each other.

**Figure 3 plants-13-03376-f003:**
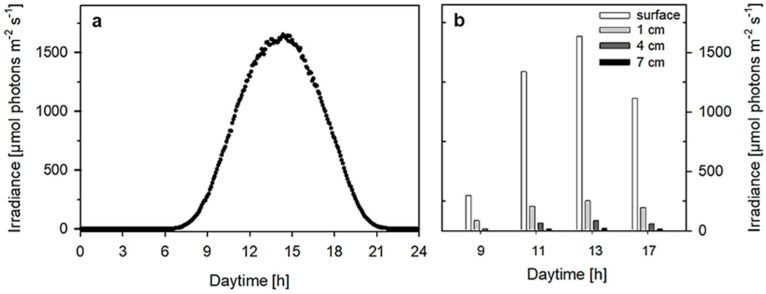
(**a**) An exemplary daily course of irradiance intensity monitored in the greenhouse during trials; (**b**) approximate irradiance levels measured in situ at the surface and 1, 4 and 7 cm deep in the culture at 9:00, 11:00, 13:00 and 17:00 h.

**Figure 4 plants-13-03376-f004:**
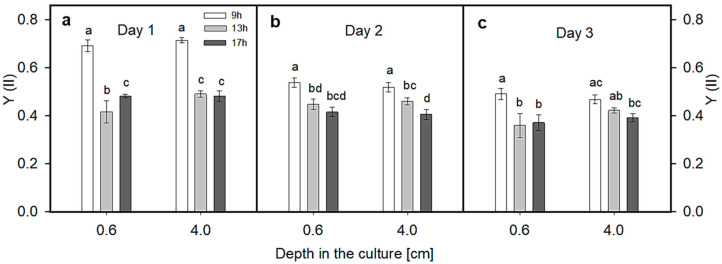
Changes in the actual photochemical yield of PSII Y(II) were measured at 9:00, 13:00 and 17:00 h on day 1 (**a**), day 2 (**b**) and day 3 (**c**) of the trial. Data were measured in situ at 0.6 and 4 cm depth in the culture using a Mini-PAM portable fluorimeter (for details, see Materials and Methods). Values are a mean ± SD of 3 measuring points in the RWP (see Figure 1b); those designated by the same letter(s) did not differ significantly from each other.

**Figure 5 plants-13-03376-f005:**
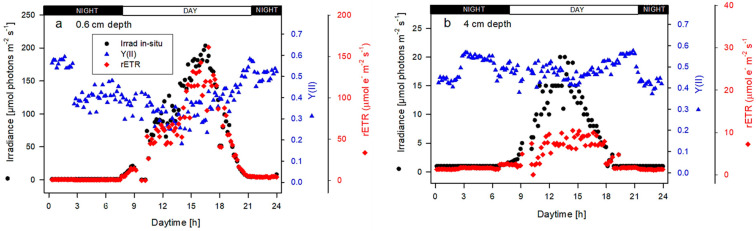
Diurnal changes in irradiance intensity (µmol photons m^−2^ s^−1^), the actual photochemical yield of PSII Y(II) and the relative electron transport rate (rETR) (µmol e^−^ m^−2^ s^−1^) measured in situ/online in the outdoor culture during the trial. Values were measured on day 3 by two Junior-PAM portable fluorimeters (for details, see Materials and Methods) at 10 min intervals with a fiberoptics and a light mini-sensor placed together at position 1 in the RWP (Figure 1) and submerged at depths of 0.6 cm (**a**) and 4 cm (**b**). The black and white bars at the upper edge of the panels indicate the day and night periods of a diurnal cycle.

**Figure 6 plants-13-03376-f006:**
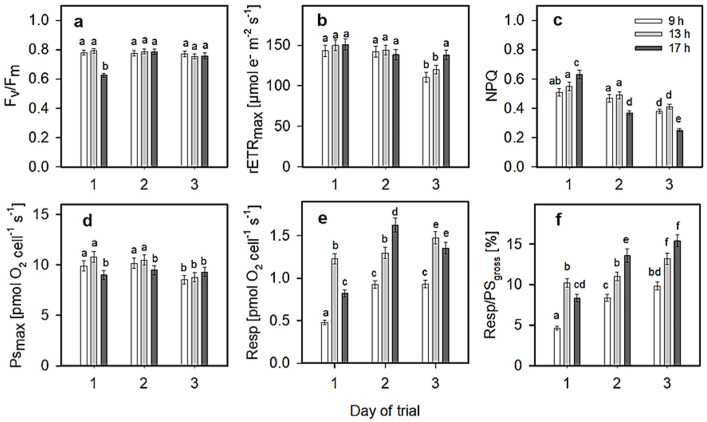
For ex situ measurements of Chl fluorescence and oxygen production, mixed microalgae samples of *Scenedesmus* were taken from outdoor cultures at 0–4 cm depth at specified daytimes (9:00, 13:00 and 17:00 h). Changes in the maximum photochemical yield of PSII (Fv/Fm) (**a**), the maxima of the relative electron transport rate (rETR_max_) (**b**), non-photochemical quenching (NPQ) (**c**), photosynthetic oxygen evolution (Pmax) (**d**), dark respiration (Resp) (**e**) and the ratio of Resp/PS_gross_ ((**f**); PS_gross_ = Resp + P_max_). The data were measured on day 1 (**a**,**d**), day 2 (**b**,**e**) and day 3 (**c**,**f**) during the three-day trial. Values were estimated from light–response curves of Chl fluorescence (PAM-2100 portable fluorimeter) and oxygen production (Oxylab+ oxygen-monitoring system), which were measured ex situ in the laboratory using the culture samples taken from the RWP (for details, see Materials and Methods). The values are presented as mean (n = 3) ± SD and those designated by the same letter did not differ significantly from each other.

**Figure 7 plants-13-03376-f007:**
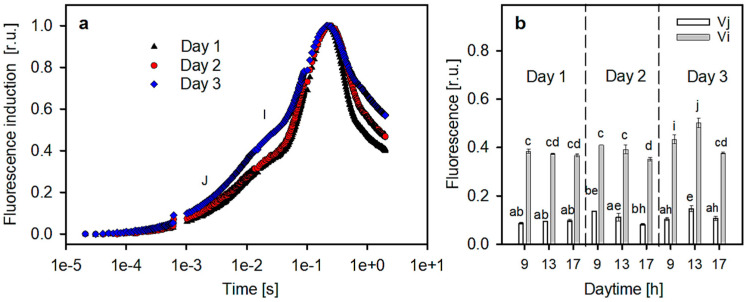
(**a**) Exemplary fast fluorescence induction kinetics measured ex situ at 13:00 h in samples taken from microalgae cultures during the three-day trial using an Aquapen AP-100 portable fluorimeter. (**b**) Changes in variables Vj and Vi were calculated from fluorescence induction curves using samples taken from cultures at particular daytimes (at 9:00, 13:00 and 17:00 h) on days 1–3. Values (for more details, see Materials and Methods) are presented as mean (n = 3) ±SD and those designated by the same letter did not differ significantly from each other.

## Data Availability

The datasets presented in this study are available from the corresponding authors upon reasonable request. The data are not publicly available without the permission of all co-authors.

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
