# Peer review of "Photosynthetic Activity Measured In Situ in Microalgae Cultures Grown in Pilot-Scale Raceway Ponds"

_plants, 2024, doi:10.3390/plants13233376_

Round 1
Reviewer 1 Report
Comments and Suggestions for Authors
The manuscript proposed by the authors is important for the field of microalgae cultivation. Indeed, the authors described the cultivation of Scenedesmus in a semi-industrial raceway pond, followed for three days, supported by a robust set of physiological analyses.
Anyway, some relevant information needs to be included, impeding the acceptance of the present version of the manuscript.
1) it needs to be clarified how the culture is cultivated. Is it batch cultivation? or continuous cultivation? The authors stated that cell biomass was around 0.6 g/L (line 138), but it is unclear what this number refers to. Is it the initial cell density? Or is the cell density kept in continuous cultivation? Authors must clarify in the text.
2) In the case of batch cultivation, what about the biomass accumulated during the three days of cultivation? This is a critical drawback and no information is provided regarding changes in terms of the biomass accumulation over time. The authors only provided indications of chlorophyll concentration (Fig. 2), which is not changing during the trial and it is difficult to interpret. If cells are actively growing, the chlorophyll concentration is expected to increase.
3) Authors do not describe why Y(PSII) is lower at day 3, but nothing is said about the changes of Y(PSII) along the three days of growth. An increase of the Y(PSII) would be expected since cells are adapted to the new conditions and are more prone to accepting photons from the sunlight. Unless the cells are in a stressed condition, but data are not clear. In fact, at day 3, the ETR max is reduced, indicating a lower photosynthetic activity because of unfavourable conditions, but this is not in agreement with the reduction of NPQ at day 3 (Figure 6). Authors should better describe in the discussion the impact of the fluctuations of these parameters along the three days of cultivation
Minors:
- line 233, "also from day toward day 3." Probably a number after day has to be added
- line 379, "and 0.015 g L-1 and commercially available mixture", one "and" has to be removed
Author Response
Comments and Suggestions for Authors
Some relevant information needs to be included, impeding the acceptance of the present version of the manuscript.
Comment 1: It needs to be clarified how the culture is cultivated. Is it batch cultivation? or continuous cultivation? The authors stated that cell biomass was around 0.6 g/L (line 138), but it is unclear what this number refers to. Is it the initial cell density? Or is the cell density kept in continuous cultivation? Authors must clarify in the text.
Response 1: The text was added, page 11, in M&M, end of par. 4.1: “The semi-continuous cultivation regime was used in this 3-day trial as when the culture dilution was done in the morning using the rate of 0.2 d-1. From the morning value of 0.6, the culture reached about 0.7 at 17:00 h.”
Comment 2: In the case of batch cultivation, what about the biomass accumulated during the three days of cultivation? This is a critical drawback and no information is provided regarding changes in terms of the biomass accumulation over time. The authors only provided indications of chlorophyll concentration (Fig. 2), which is not changing during the trial and it is difficult to interpret. If cells are actively growing, the chlorophyll concentration is expected to increase.
Response 2: This comment is partially answered in Response 2 above as the culture was partially diluted every day. The text was added – page 4, l. 148-151: “Actually, the Chl concentration varied on day 1 and on day 2 it slightly (statistically) increased from the morning till the afternoon. On day 3, it did not change since the culture was growing slowly.
Comment 3: Authors do not describe why Y(PSII) is lower at day 3, but nothing is said about the changes of Y(PSII) along the three days of growth. An increase of the Y(PSII) would be expected since cells are adapted to the new conditions and are more prone to accepting photons from the sunlight. Unless the cells are in a stressed condition, but data are not clear. In fact, at day 3, the ETR max is reduced, indicating a lower photosynthetic activity because of unfavourable conditions, but this is not in agreement with the reduction of NPQ at day 3 (Figure 6). Authors should better describe in the discussion the impact of the fluctuations of these parameters along the three days of cultivation.
Response 3: Judging from the Fv/Fm data (Fig. 6a), one would not expect a decrease in the Y(II) with the increase of the cell density and thus an increasing self-shading which would reduce the average light to cells. Nevertheless, the Y(II) values decreased during the cultivation trial. The culture was set up on day 1, the high morning value of Y(II) was probably induced due to that. On day 2 and namely day 3 the afternoon decrease of Y(II) can be probably ascribed to the synergistic effect of higher irradiance, excessive temperature and DO concentration. A similar trend was noted in the rise of the ratio of respiration/photosynthesis measured ex‑situ (Fig. 6f). Surprisingly, NPQ showed a tendency to decrease (Fig. 6c). The reason for the decrease in NPQ during the cultivation trial is not clear, however, a stress effect of excessive temperature and dissolved oxygen concentration on the capacity to dissipate energy via NPQ cannot be ruled out, as well as energy dissipation in some other mechanisms (futile cycling in photosystems, photorespiration).
Minors:
Comment 4: - line 233, "also from day toward day 3." Probably a number after day has to be added
Response 4: The reviewer was correct, now line 237 was revised to: … from day 1 to day 3
Comment 5: - line 379, "and 0.015 g L-1 and commercially available mixture", one "and" has to be removed.
Response 5: Now line 385, corrected to: … and 0.015 g L-1 commercially available mixture of micronutrients @karentol …
Reviewer 2 Report
Comments and Suggestions for Authors
The manuscript written by Jiří Masojídek et al. and titled ‘Photosynthetic activity measured in-situ in microalgae cultures grown in pilot-scale raceway ponds’ makes an attempt to compare the states of photosynthetic apparatus in culture of green alga Scenedesmus grown in an open raceway pond. In general, the main idea of the work is not scientifically important because it was widely studied in a range of microalgae species including Scenedesmu and the data obtained are not different significantly from that of known, besides growth in the pilot-scale raceway ponds. In addition, the authors are misunderstanding the data obtained by chlorophyll fluorescence approaches, including calculations, which led to incorrect conclusions in some points. Thus, I think that the manuscript in the current view required serious corrections.
My comments, questions, and main incorrect places I list below.
The very important point during measurement of Y(II) is the wavelength of the actinic light, which was indicated (blue), and its intensity, which is not indicated, but must be. This is because Y(II) is a photochemical efficiency of PSII adapted to light (in contrast to Fv/Fm, which is the maximum potential of PSII for photochemistry). Therefore, under higher light intensity Y(II) is decreased and vice versa.
Value of Y(II) near 0.5 is usually used, which is close to that in Fig 4, but why were the values of 0.71 observed for Day 1(9.00)? How can the authors explain it?
The authors write that ‘the Y(II) values measured at 0.6-cm 176 and 4-cm depths were lower – between 0.31-0.51 due to higher ambient irradiance penetrating the culture (L177)’, however above (L158 and Fig3) the significant difference was indicated even between 1 and cm depths, in addition see L204, and see also L327 in the Discussion. At the same time in L180 the authors write ‘that penetrating irradiance intensity into the culture was similar’. How do the authors comment on this discrepancy?
From which data or articles (there are no reff) did the authors conclude about higher PQ pool reduction (L178)? In particle support of this they should detect the decrease in the Area parameter, but they did not (see Fig 7a).
The chlorophyll fluorescence quenching can not be accumulated (L181), because this is a process. Should be corrected.
I doubt the correct use of rETR calculation because the indicated equitation is incomplete (see also Methods). According to the Manual of the PAM used it should be as follows – ETR= Y(II)*PAR * 0.84 * 0.5 (p32), or ETR = PAR · ETR-Factor · Y(II), as mentioned in Manuals to other PAMs of Walz. Thus the authors should completely revise the data including Fig 5 and 6.
I do not understand why the authors indicate data of OJIP. I would say that according to them the state of PSII in all variants was the same. Higher level at J should be the result of short dark acclimation of cells rather than with the decrease in PQ reduction. Usually not less than 20 min is required for dark adaptation. The same see in L350-352. Correct it.
In the Discussion the authors indicated the absolute misunderstanding of mechanisms of chlorophyll fluorescence induction. I would recommend them to read any high level articles about it, for example doi/10.1104/pp.15.01935 and 10.1016/s0005-2728(99)00047-x. Surprisingly, both of these articles are not in the References.
Based on this – NPQ is not only dissipation of energy (L274), but also distribution of the absorbed energy between photosystems (state transitions (ST); more important that qE is fast-development rather than fast-relaxing (L275). In addition, for its induction the accumulation of Lhscrs3 and PsbS proteins as well as xanthophyll should be occurred; NPQ form →NPQ component;
The authors write ‘since NPQ remains still active during the transition of the cells from the light-saturated layer to the light-limited part of the culture layer (L291), but above they cooled NPQ as fast-relaxing (L275). Which is correct? It is assumed that qE develops and relaxes in seconds range up to 1 min. The authors should correct their idea.
Carotenoids are the second participants in NPQ induction. The first one are Lhcsr3 and PsbS proteins. In addition, all NPQ mechanisms are based on the decrease of lumenal pH. The authors should correct this part. See also L361-363. Correct it.
Y(NO) is nonregulated NPQ usually correlated with forming of Qb closed reaction centers or with PSII photodamage, but not with energy dissipation. Dissipation and distribution of absorbed energy occur via Y(NPQ). This paragraph should be completely rewritten.
Comments on the Quality of English LanguageMinor corrections are requared
Author Response
Comments and Suggestions for Authors
Comment 1: In general, the main idea of the work is not scientifically important because it was widely studied in a range of microalgae species including Scenedesmus and the data obtained are not different significantly from that of known, besides growth in the pilot-scale raceway ponds. In addition, the authors are misunderstanding the data obtained by chlorophyll fluorescence approaches, including calculations, which led to incorrect conclusions in some points. Thus, I think that the manuscript in the current view required serious corrections.
Response 1: To respond to this comment we need a longer explanation. We wish to emphasize that the main purpose of this study was to employ chlorophyll fluorescence for monitoring potentially unfavourable conditions in microalgae cultures grown in open ponds. To the best of our knowledge, not much information has been available on chlorophyll fluorescence changes at different culture depths. Indeed, most measurements were carried out by monitoring fluorescence yields on the surface of cultures, or by withdrawing samples from cultures and transferring them to the lab. Chlorophyll fluorescence, due to its fast response and sensitivity, is a valid tool for monitoring and understanding how culture performance changes within the culture depth.
Chl fluorescence data and calculations were thoroughly considered comparing both in-situ and ex-situ measurements according to our long-time experience and literature (in the list of references: Torzillo et al. 1998, Figueroa et al. 2013, Malapascua et al. 2014, Jerez et al. 2016, Masojídek et al. 2021, Rearte et al. 2021). Nevertheless, at present the interpretation of Chl fluorescence variables is well elaborated as shown in the abovementioned references. We did not find any serious failures in calculations and interpretations. However, it is important to point out that with microalgae cultures, particular care is needed when evaluating fluorescence data, for example, those of ETR.
There are several points in the text of part 5 - Conclusions (lines 540-560) which contribute to the importance of results. (i) Generally, photosynthesis-measuring techniques are reliable for culture monitoring as they reflect the physiological status and growth of microalgae in large-scale photobioreactors; (ii) The correlation between culture photosynthetic activity and growth can be used to optimize the cultivation regime of microalgae cultures in large-scale units. (iii) The presented data provide important information on the physiological status of cells at different culture depths. Online photosynthesis-measuring techniques (Chl fluorescence) showed that the ‘real’ photic zone in the open ponds contributing to growth is relatively thin, about 1 cm; in this case, less than 10% of the culture depth. (iv) Understanding the photosynthesis performance of cells at different depths is in our opinion the key point to optimize mixing and thus biomass yields.
Comment 2: The very important point during measurement of Y(II) is the wavelength of the actinic light, which was indicated (blue), and its intensity, which is not indicated, but must be. This is because Y(II) is a photochemical efficiency of PSII adapted to light (in contrast to Fv/Fm, which is the maximum potential of PSII for photochemistry). Therefore, under higher light intensity Y(II) is decreased and vice versa.
Response 2. We believe these are concepts that are clear to us; the higher the actinic light the lower the fluorescence yield (effective quantum yield). In these trials, we combined several instruments and techniques to obtain a reliable picture. In outdoor trials, solar light was used as the actinic and we combined in-situ and ex-situ measurements using various fluorimeters which employ various actinic light sources: either blue 450-470 nm (Junior-PAM) or red 650 nm (Mini-PAM, PAM-2100, Aquapen AP‑100), or also white (PAM-2100). It is important to emphasize that the intensity of measuring light to monitor Chl fluorescence variables has to be subsaturating for photosynthesis.
Comment 3: Value of Y(II) near 0.5 is usually used, which is close to that in Fig 4, but why were the values of 0.71 observed for Day 1(9.00)? How can the authors explain it?
Response 3. On the first day starting the trial after dilution, the culture was relaxed at the time of the measurements; thus Y(II) value was higher compared to the measurements at 9:00 h on days 2 and 3. At midday and in the afternoon, the Y(II) values were lower due to PSII reduction (reaction center closure) at higher irradiance of the culture.
Comment 4: The authors write that ‘the Y(II) values measured at 0.6-cm and 4-cm depths were lower – between 0.31-0.51 due to higher ambient irradiance penetrating the culture (L177)’, however above (L158 and Fig3) the significant difference was indicated even between 1 and 4 cm depths, in addition, see L204, and see also L327 in the Discussion. At the same time in L180 the authors write ‘that penetrating irradiance intensity into the culture was similar’. How do the authors Comment on this discrepancy?
Response 4: In this comment, there is probably some misunderstanding of the text. There was a dramatic difference between the irradiance intensity at the surface and 1-cm culture depth (80-90 % decrease) and then more than another 50% decrease between 1-cm and 4-cm depths. The course of the Y(II) values along the day shows an antiparallel behavior with changes in solar light intensity. Due to the light extinction, one might expect that the Y(II) value would be higher in deeper layers (i.e. a recovery via PSII reoxidation) but the cultures are mixed and cells are probably moved from upper, more active (illuminated) layers.
To avoid any misinterpretation these parts of the text were revised (l.166-170):” The data showed that there was a dramatic difference between the irradiance intensity at the surface and 1-cm culture depth (80-90 % decrease) and then more than 50% decrease between 1-cm and 4-cm depths.”
Then another discussed part was revised (l.183-188): ”At 13:00 h and 17:00 h, the Y(II) values measured at 0.6-cm and 4-cm depths were lower than in the morning – between 0.31-0.51 due to prolonged exposure to daily irradiance which caused downregulation of PSII. Cells, even at 4 cm depth preserved some form of quenching developed on the surface which could further reduce the ETR capacity, even when transferred to lower irradiance ( 4 cm layer), particularly at midday when irradiance intensity experienced by cells on the top layer is higher.”
Comment 5: From which data or articles (there are no ref) did the authors conclude about higher PQ pool reduction (L178)? In particle support of this they should detect the decrease in the Area parameter, but they did not (see Fig 7a).
Response 5. This conclusion was deducted from data elaboration of the Kautsky curve, namely the Vj and Vi variables corresponding to the rise of the J and I steps indicating the overreduction of plastoquinone electron acceptors (Strasser et al. 2004).
Comment 6: The chlorophyll fluorescence quenching can not be accumulated (L181), because this is a process. Should be corrected.
Response 6. We agree with the reviewer, instead we used the term ‘fluorescence quenching developed’.
Comment 7: I doubt the correct use of rETR calculation because the indicated equitation is incomplete (see also Methods). According to the Manual of the PAM used it should be as follows – ETR= Y(II)*PAR * 0.84 * 0.5 (p32), or ETR = PAR · ETR-Factor · Y(II), as mentioned in Manuals to other PAMs of Walz. Thus the authors should completely revise the data including Fig 5 and 6.
Response 7. This is a long-debated question among people using fluorimeters to monitor the photosynthetic performance of plants and microalgae cultures. Indeed, fluorimeters were designed to measure fluorescence on plant leaves. The equation to calculate the relative electron transport rate, ETR= Y(II) * PAR * 0.84 * 0.5 was elaborated for leaves using PAM fluorimeters [ref. Schreiber U. 2004, Chapter 11. Pulse-Amplitude-Modulation (PAM) Fluorometry and Saturation Pulse Method: An Overview. In Chlorophyll a Fluorescence: A Signature of Photosynthesis, pp. 279-319]. .
In this equation, 0.84 means that the average leaf absorbs about 84% of incident PAR and 0.5 is a number based on equal electron transport rates through PSII and PSI. But, in diluted microalgae suspensions (like that used in our experiments), the light absorption is about 100% and the electron transport rate through photosystems is equal. Then, we can simplify the equation to rETR = Y(II) x PAR which is sufficient for comparative purposes. Therefore, this variable is called rETR. Such an approach has been frequently used in literature (e.g. Hofstraat et al. 1994, Ralph and Gademann 2005, White al. 2011, Malapascua et al. 2014).
Comment 8: I do not understand why the authors indicate data of OJIP. I would say that according to them the state of PSII in all variants was the same. Higher level at J should be the result of short dark acclimation of cells rather than with the decrease in PQ reduction. Usually not less than 20 min is required for dark adaptation. The same see in L350-352. Correct it.
Response 8. Based on our measurements and experience with microalgae, 10-15 min dark adaptation period is sufficient for outdoor cultures exposed to high irradiance. This period allows the photosystem II (PSII) reaction centers to open (oxidise), enabling accurate measurements of variables like the maximum quantum yield (Fv/Fm).
As concerns the interpretation of fast fluorescence induction curves, we used the widely accepted expertise of Strasser and co-workers based on a long experience in this field which is adequate for the present MS. According to this, the J inflexion indicates the equilibrium between QA reduction and QA reoxidation by the PQ pool (Strasser et al. 2004, Goltsev et al. 2016). An accumulation of QA- in the J step was also demonstrated by fluorescence increase in this step after the addition of DCMU (Strasser et al. 1995 Photochem. Photobiol. 61).
The data calculated from the curves of fast fluorescence induction kinetics showed that on day 3 the electron transport on the acceptor side of the PSII complex was slightly affected, just delayed probably due to the partial overreduction of QA. In the text, we only say (l. 370-377): “The kinetics of fast fluorescence induction did not show any deep disturbance in the PSII complex (Figure 7). It suggests that the electron transport on the acceptor side of the PSII complex was slightly affected, just delayed probably due to the partial overreduction of QA. A certain increase of the Vj and Vi values was observed on day 3 which might suggest that the electron transport in the cell was slowed down.”
Comment 9: In the Discussion, the authors indicated the absolute misunderstanding of mechanisms of chlorophyll fluorescence induction. I would recommend them to read any high level articles about it, for example doi/10.1104/pp.15.01935 and 10.1016/s0005-2728(99)00047-x. Surprisingly, both of these articles are not in the References.
Response 9. We are sorry not to discuss the abovementioned references. At this point, there is probably some misunderstanding. As far as we studied the article by Lazár 1999, he widely cited numerous articles by Strasser and co-workers agreeing with the interpretation of their OJIP fluorescence induction kinetics and the J and I points as it was cited in this MS and explained in Response 8 above.
Comment 10: Based on this – NPQ is not only dissipation of energy (L274), but also distribution of the absorbed energy between photosystems (state transitions (ST); more important that qE is fast-development rather than fast-relaxing (L275). In addition, for its induction the accumulation of Lhscrs3 and PsbS proteins as well as xanthophyll should be occurred; NPQ form →NPQ component;
Response 10. We agree with the comment as the text was not completely accurate. The discussion part was partly revised (l. 274-286): “Mechanisms collectively referred to as non-photochemical quenching (NPQ) are developed in higher plants [Demmig-Adams and Adams 1992] as well as microalgae [Masojidek et al. 1999] to dissipate excess energy as heat, or to distribute excitation energy (state transition) between the two photosystems, thus protecting the photosynthetic apparatus from such damage [Krause and Jahns 2004]. The most significant component of the NPQ is the energy-dependent quenching qE. According to current understanding, qE is developed based on the synergistic action of the lumen pH, xanthophyll interconversion and the conformational changes in PSII antenna proteins Lhcb and PsbS protein [Gilmore et al. 1998]. In the culture of microalgae, like in higher plants, qE is activated when the amount of incoming light energy exceeds the capacity of electron sinks. The participation of another component, state transition quenching qT cannot be ruled out. The threshold light level inducing this process is lowered under unfavourable conditions, such as high salinity, nutrient deficiency, or suboptimal temperature which further limit photosynthesis [Krause and Jahns 2004].”
Comment 11: The authors write ‘since NPQ remains still active during the transition of the cells from the light-saturated layer to the light-limited part of the culture layer (L291), but above they cooled NPQ as fast-relaxing (L275). Which is correct? It is assumed that qE develops and relaxes in seconds range up to 1 min. The authors should correct their idea.
Response 11. As above in Response 10, a part of the Discussion text was revised and the term ‘fast-relaxing’ was eliminated to avoid controversy.
Comment 12: Carotenoids are the second participants in NPQ induction. The first one are Lhcsr3 and PsbS proteins. In addition, all NPQ mechanisms are based on the decrease of lumenal pH. The authors should correct this part. See also L361-363. Correct it.
Response 12. The referee is right, we have revised these parts as mentioned above in Response 10.
Comment 13: Y(NO) is nonregulated NPQ usually correlated with forming of Qb closed reaction centers or with PSII photodamage, but not with energy dissipation. Dissipation and distribution of absorbed energy occur via Y(NPQ). This paragraph should be completely rewritten.
Response 13. The last paragraph of the Discussion was completely rewritten as follows (l.390-403) “In this work, non-photochemical quenching can be calculated as NPQ = (Fm‑Fm’)/Fm’ [Baker 2008]. In some reports, NPQ is also expressed as the ratio of the two components, i.e. Φ(NPQ)/Φ(NO) [Kramer et al. 2004b, Hendrickson et al. 2004], or Y(NPQ)/Y(NO) [Klughammer and Schreiber 2008]. These two components of non-photochemical quenching can be used to estimate the fluxes of excitation energy into competing pathways. Y(NPQ) corresponds to the fraction of energy dissipated in the form of heat via the down-regulatory photoprotective mechanism with the participation of carotenoids while Y(NO) according to the above-cited authors reflects the fraction of energy that is passively dissipated in the form of heat and fluorescence, mainly due to closed (PQ reduced) PSII reaction centers. The sum of the two components, i.e. Y(NPQ)+Y(NO) plus the actual quantum yield Y(II) is 1. The values of NPQ lower than 1 indicate that non-regulated loss of energy Y(NO) is dominating compared to the energy dissipation mechanism Y(NPQ) related to carotenoids. High values of Y(NO) indicate an inability of the microalgae cells to protect themselves against photodamage by excess radiation (Kramer et al. 2004, Hendrickson et al. 2004).
Reviewer 3 Report
Comments and Suggestions for Authors
The manuscript has focused on the research of Photosynthetic activity measured in-situ in microalgae cultures grown in pilot-scale raceway ponds. Please see below my observations:
1. The explanation in the literature review section have depth. It showing the research trend in the related field, and critically review the previous related studies and reveal the knowledge gaps and inconsistencies in the literature. The authors relate the objectives of the study but I don’t see where the authors present the research novelty and research significance of the study in contrast with other similar studies.
2. Please use Scenedesmus sp in all the manuscript. there are places in the manuscript where only Scenedesmus is used.
3. I suggest to the authors to use the term photobioreactor instead of bioreactor.
4. Please use the same font size in all the Chapters of the paper.
Author Response
Comments and Suggestions for Authors
The manuscript has focused on the research of Photosynthetic activity measured in-situ in microalgae cultures grown in pilot-scale raceway ponds. Please see below my observations:
Comment 1: The explanation in the literature review section have depth. It showing the research trend in the related field, and critically review the previous related studies and reveal the knowledge gaps and inconsistencies in the literature. The authors relate the objectives of the study but I don’t see where the authors present the research novelty and research significance of the study in contrast with other similar studies.
Response 1: First of all, in addition to measurements carried out in the past, we compared the Y(II) and rETR at two culture depths. Indeed, chlorophyll measurements have usually been carried out on the surface of the culture (which could be satisfactory for leaves) but the cells of microalgae are subjected to mixing and light attenuation along the culture layer which strongly changes the light intensity and thus the fluorescence. Because of rapid, complex and well-elaborated measurements, fluorescence represents an ideal tool to assess culture performance.
As responded to Reviewer 2, there are several conclusions which contribute to the novelty of the research results: (i) The correlation between culture photosynthetic activity and growth can be used to optimize the cultivation regime of microalgae cultures in large-scale units. (ii) Understanding the photosynthesis performance of cells at different depths is – in our opinion – the key point to optimize mixing and thus biomass yields. Online photosynthesis-measuring techniques (Chl fluorescence) provide important information on the physiological status of cells along the culture column. The presented data revealed that the ‘real’ photic zone in the open ponds contributing to growth is relatively thin, about 1 cm; in this case, less than 10% of the culture depth.
Comment 2. Please use Scenedesmus sp in all the manuscript; there are places in the manuscript where only Scenedesmus is used.
Response 2: As we used only one microalgal strain, i.e. Scenedesmus sp., it is sufficient – in our opinion – to use a one-word abbreviation Scenedesmus to simplify the text.
Comment 3. I suggest to the authors to use the term photobioreactor instead of bioreactor.
Response 3: In the MS text the term bioreactor was replaced by photobioreactor.
Comment 4. Please use the same font size in all the Chapters of the paper.
Response 4: The font size and formatting in various parts of the text are given in the template form of the journal Plants.Konec formuláře
Round 2
Reviewer 1 Report
Comments and Suggestions for Authors
Thanks for the adjustements.
The paper can be accepted for publication. Congratulations
Author Response

(The authors gave the same response as above.)

Reviewer 2 Report
Comments and Suggestions for Authors
The manuscript of Jiří Masojídek et al. titled ‘Photosynthetic activity measured in-situ in microalgae cultures grown in pilot-scale raceway ponds’ was reviewed in the second round. Unfortunately, the authors almost did not correct the text and left many scientific errors in it. Based on this I can not recommend the acceptance of the manuscript for publication again, but try to require major revision.
I think that I should not list all my comments again because the authors can see my previous report. I only indicate some important moments.
I actually think that the phrase ‘…relax the photosynthetic apparatus…’ (Line 35 and therein) is not scientific suitable.
The NPQ (quenching) can not be accumulated (L193) because this is a process induced by light like a reflection in the mirror. But the algal cells can be adapted to different light conditions, for example, by synthesis of Lhcsr3 and PsbS proteins, carotenoids, and so on. In addition, the light intensity near 20 mkmol (L183) is not enough to induce any NPQ in 4 cm deep.
I am sure that Fig 4A has a mistake and the explanation like ‘the culture was relaxed at the time of the measurements’ is out of science. Can the authors provide the induction curves measured for cultures at 9 am in 1, 2, and 3 days and used for Y(II) calculations? The analysis can help to understand the reasons. It is obvious now that these are not light intensity and deep of the measurement.
I disagree with the authors about the calculation of rETR. Probably they can not use a coefficient of 0.84 if they want, in spite of the fact that other authors used (see doi.org/10.1007/s10811-023-02915-2; doi.org/10.1016/j.algal.2018.05.018; DOI 10.32615/ps.2021.054), but anyway they should take in the amount the distribution between PSII and PSI, i.e. to use a factor of 0.5. Or make the calculations of Sigma(II), the wavelength-dependent absorption cross-section of PS II (see the manual: To obtain reasonable fluorescence-based estimates of electron transport rates in suspensions, PamWin-3 uses the approach introduced by Schreiber et al. (2011) which permits calculation of quanta absorbed by PS II, PAR(II), from incident PAR and Sigma(II), the wavelength-dependent absorption cross-section of PS II.). Thus the authors should recalculate all rETR data presented in the manuscript.
The authors do not discuss the Lhcsr3 protein in terms of NPQ induction, but should. For example, here – Line 297.
The culture was set up on day 1 which might induce the high morning value (L327). See above.
(L 335)The reason for the decrease in NPQ during the cultivation trial is not clear. – This is not correct. For example, the decrease may occur during long-term adaptation of the photosynthetic apparatus to light and growth condition through protein (Lhcsr3, PsbS) and carotenoids accumulation, changes in antenna size, the PSII to PSI ratios, the thylakoid-related architecture, and others. The authors should be more accurate.
(L426-427). The value of any of three components (Y(II), Y(NPQ), and Y(NO), if we talk about the quantum yields, are always less than 1. Thus, I think that the sentence ‘The values of NPQ lower than 1 indicate that non-regulated loss of energy Y(NO) is dominating…’ is absolutely incorrect.
L429. I repeat again, the Lhcsr3 protein is much more important for NPQ induction in green algae than carotenoids. Why do the authors continue to ignore it? They should add information about this protein.
Author Response
Reviewer 2 – second round of comments
The manuscript of Jiří Masojídek et al. titled ‘Photosynthetic activity measured in-situ in microalgae cultures grown in pilot-scale raceway ponds’ was reviewed in the second round. Unfortunately, the authors almost did not correct the text and left many scientific errors in it. Based on this I cannot recommend the acceptance of the manuscript for publication again, but try to require major revision.
Response to the introductory part of the report: We thoroughly revised the MS text in the first round of comments. Now, we have been surprised to receive another bunch of comments. Again in this response, we answer substantial comments in detail and make necessary changes in the text.
In this point, we wish to emphasize that the novelty of the work was the use of chlorophyll fluorescence to monitor potentially unfavorable conditions in large-scale open pond cultures of Scenedesmus comparing photosynthetic performance in various layers of the culture volume. Chl fluorescence was employed by us in several papers (see list of references), but here we investigated the performance of the culture at two different and crutial depths, and in our opinion as well as two other referees the results deserve publication.
On the statement of referee 2 “In addition, the authors are misunderstanding the data obtained by chlorophyll fluorescence approaches, including calculations, which led to incorrect conclusions in some points,”, we respond in the points below. For us, such a comment sounds very thought-provoking since in calculating rETR and NPQ, questioned by the referee, we followed formulas published by outstanding authors – Hofstraat et al. 1994, Krammer et al. 2004, Ralph and Gademann 2005, White al. 2011, Cosgrove and Borowitzka 2011, among others – and in addition in the articles published by our team in Q1 journals (in the list of references Torzillo et al. 1998, Figueroa et al. 2013, Malapascua et al. 2014, Jerez et al. 2016, Rearte et al. 2021, Masojídek et al. 2021, 2022, 2023) on photosynthesis monitoring in microalgae mass cultures by chlorophyll fluorescence. Nevertheless, some of the calculations differ when used for microalgae vs. higher plants.
Comment 1: I think that I should not list all my comments again because the authors can see my previous report. I only indicate some important moments.
Response 1: Again, we have thoroughly studied all the comments of the previous report, as well as in this one, responding to them in detail and noteworthy changes can be found in the revised MS.
Comment 2: I actually think that the phrase ‘…relax the photosynthetic apparatus…’ (Line 35 and therein) is not scientific suitable.
Response 2: Well, we rearticulated the sentence in the abstract to: “ … (ii) the mechanism of non-photochemical quenching may not be fast enough to respond when the cells move from the surface to the deeper layers;…
Comment 3: The NPQ (quenching) can not be accumulated (L193) because this is a process induced by light like a reflection in the mirror. But the algal cells can be adapted to different light conditions, for example, by synthesis of Lhcsr3 and PsbS proteins, carotenoids, and so on. In addition, the light intensity near 20 mkmol (L183) is not enough to induce any NPQ in 4 cm deep.
Response 3: We agree an unsuitable expression was used – NPQ cannot be accumulated but it is well know that different types of quenching take different times to relax. Various NPQ forms are distinguished according to the relaxing time. However, the sentence in the last paragraph of the Introduction was clarified/rephrased to: “Surprisingly, when the cells were moved to the bottom layer, i.e. low light conditions by mixing, the photosynthetic apparatus continued to operate at a lower efficiency as a result of some form of quenching not fully diminished which could reduce photochemical performance, particularly at midday when irradiance was higher.”
We do not say that the irradiance of 20 micromol photons/m2s induces some quenching but it is stated: “Cells, even at 4- cm depth preserved some form of quenching developed on the surface which could further reduce the ETR capacity, particularly at midday when irradiance intensity is higher.” The reviewer should consider that NPQ requires a certain time to get induced or to get relaxed. The light/dark cycle at which cells are circulated by mixing may not facilitate a complete relaxation of NPQ. In addition, NPQ was defined semi-empirically and is calculated as the ratio of fluorescence yields Fm-Fm’/Fm’, similar to the actual photochemical yield Y(II). It is not difficult to figure out that rearrangement of PSII antenna chlorophyll-proteins or the xanthophyll cycle pigments cannot be completed once the cells are promptly transferred from low to high light and vice versa.
Comment 4: I am sure that Fig 4A has a mistake and the explanation like ‘the culture was relaxed at the time of the measurements’ is out of science. Can the authors provide the induction curves measured for cultures at 9 am in 1, 2, and 3 days and used for Y(II) calculations? The analysis can help to understand the reasons. It is obvious now that these are not light intensity and deep of the measurement.
Response 4: We disagree with such a comment as there is probably some kind of misunderstanding. The values of Y(II) were measured in-situ, 0.6 and 4 cm deep in the culture column using a portable modulated fluorimeter Mini-PAM, not using fluorescence induction. Moreover, the data are a mean ± SD recorded at certain daytime in three measuring points in the RWP. The explanation can be very simple. During the preceding days, light intensity was lower due to culture density and thus Y(II) was higher. Once we started with measurements there was a steep rise in light irradiance and this caused a reduction in Y(II). However, we thank the referee for mentioning this point which was promptly explained in the text.
Comment 5: I disagree with the authors about the calculation of rETR. Probably they can not use a coefficient of 0.84 if they want, in spite of the fact that other authors used (see doi.org/10.1007/s10811-023-02915-2; doi.org/10.1016/j.algal.2018.05.018; DOI 10.32615/ps.2021.054), but anyway they should take in the amount the distribution between PSII and PSI, i.e. to use a factor of 0.5. Or make the calculations of Sigma(II), the wavelength-dependent absorption cross-section of PS II (see the manual: To obtain reasonable fluorescence-based estimates of electron transport rates in suspensions, PamWin-3 uses the approach introduced by Schreiber et al. (2011) which permits calculation of quanta absorbed by PS II, PAR(II), from incident PAR and Sigma(II), the wavelength-dependent absorption cross-section of PS II.). Thus the authors should recalculate all rETR data presented in the manuscript.
Response 5: Again, this is a long-debated question among people using fluorimeters to monitor and evaluate the photosynthetic performance of plants and microalgae cultures. We emphasize that fluorimeters were designed to measure fluorescence on plant leaves but in this work, we use microalgae aquacultures which require experimentally different approach. In this point of how to calculate rETR, we have consistently different opinions as already discussed in the first round of comments. Our approach is supported by numerous reports (e.g. Hofstraat et al. 1994, Ralph and Gademann 2005, Cosgrove and Borowitzka 2006, White al. 2011, Cosgrove and Borowitzka 2011, Malapascua et al. 2014). Accordingly in this MS, the equation to calculate was simplified to rETR [= Y(II) x PAR] which is sufficient and fair for comparative purposes during diurnal measurements of mass microalgae cultures. It was reported to calculate the relative electron transport rate through PSII, either rETR = Fq′/Fm′ × EPAR, or alternatively Fq′/Fm′ × EPAR × 0.5 (Cosgrove and Borowitzka 2011 Chapter 1. Chlorophyll Fluorescence Terminology: An Introduction. D.J. Suggett et al., eds., Chlorophyll a Fluorescence in Aquatic Sciences: Methods and Applications, Springer Science+Business Media). If we omit the factor of 0.84 (as the absorption coefficient of incident PAR by leaf) as admitted by the referee, then the number of 0.5 is just a multiplication factor based on the assumption that 50% of the absorbed quanta are distributed to PSII. In practice, it can also be omitted as it does not influence the comparative purpose of rETR. We hope that now the disputed point has been clarified.
Comment 6: The authors do not discuss the Lhcsr3 protein in terms of NPQ induction, but should. For example, here – Line 297.
Response 6: At this point, we followed the suggestion of the referee and included new text in the first and second paragraphs of the discussion:
“Recently, the NPQ scenario was demonstrated in higher plants as NPQ = ΔpH + PsbS + LHCII (Ruban 2016). It was proposed that NPQ is triggered by ΔpH either directly by protonation of antenna components or indirectly, by the activity of the xanthophyll cycle. LHCII complexes CP26 and CP29 can accept protons attaining high levels of quenching and are enriched in xanthophyll cycle carotenoids. A component that plays a crucial role in enabling the rapidly reversible component of NPQ, qE, is the PsbS protein which acts like a switch that is also triggered by ΔpH and is localized closer to the LHCII antenna to prompt it into a quenching state or make it sensitive to protonation (Ruban and Johnson 2012).
In the culture of microalgae, like in higher plants, qE is activated when the amount of incoming light energy exceeds the capacity of electron sinks. The participation of another component, state transition quenching qT cannot be ruled out. The threshold light level inducing this process is lowered under unfavourable conditions, such as high salinity, nutrient deficiency, or suboptimal temperature which further limit photosynthesis [29]. While the requirements for NPQ development in green microalgae, i.e. the xanthophyll cycle, the proton gradient and LHCII aggregation are comparable to those of higher plants, there also exist important differences (Goss and Lepetit 2015). The most prominent difference in the requirements of NPQ between higher plants and green microalgae is that in several strains the PsbS protein is not present and is replaced by the so-called Lhcsr proteins, ancient members of the LHC protein superfamily. It was shown that the Lhcsr3 protein of Chlamydomonas reinhardtii has features which are comparable to those of PsbS (Bonente et al., 2011). Like this protein, Lhcsr3 is also sensitive to an increase in the proton concentration during the formation of the proton gradient (Peers et al. 2009; Bonente et al. 2011; Gerotto and Morosinotto 2013).
Comment 7: The culture was set up on day 1 which might induce the high morning value (L327). See above.
Response 7: See Response 4
Comment 7: (L 335)The reason for the decrease in NPQ during the cultivation trial is not clear. – This is not correct. For example, the decrease may occur during long-term adaptation of the photosynthetic apparatus to light and growth condition through protein (Lhcsr3, PsbS) and carotenoids accumulation, changes in antenna size, the PSII to PSI ratios, the thylakoid-related architecture, and others. The authors should be more accurate.
Response 7: One sentence was added in the discussion part (p. 9).”It may occur during long-term adaptation of the photosynthetic apparatus to light and growth conditions through antenna proteins and carotenoid accumulation, changes in antenna size, the PSII to PSI ratios, the thylakoid-related architecture, and others (Ruban 2016).”
Comment 8: (L426-427). The value of any of three components (Y(II), Y(NPQ), and Y(NO), if we talk about the quantum yields, are always less than 1. Thus, I think that the sentence ‘The values of NPQ lower than 1 indicate that non-regulated loss of energy Y(NO) is dominating…’ is absolutely incorrect.
Response 8: This part of the text was confusing and it was revised to: “The sum of the two components, i.e. Y(NPQ)+Y(NO) plus the actual photochemical yield Y(II) is 1. The lower values of NPQ indicate that non-regulated loss of energy, Y(NO) is dominating compared to the photoregulated energy dissipation mechanism Y(NPQ) related to carotenoids [42,44].
Comment 9: L429. I repeat again, the Lhcsr3 protein is much more important for NPQ induction in green algae than carotenoids. Why do the authors continue to ignore it? They should add information about this protein.
Response 9: Following the comment of the referee, the text was added to the discussion part as mentioned in Response 6.